# Genetic Background and Kinetics Define Wound Bed Extracellular Vesicles in a Mouse Model of Cutaneous Injury

**DOI:** 10.3390/ijms22073551

**Published:** 2021-03-29

**Authors:** Jin Qian, Dong Jun Park, Sophia Perrott, Parth Patel, Brian P. Eliceiri

**Affiliations:** 1Division of Trauma, Department of Surgery, UC San Diego Health Sciences, 212 Dickinson Street, MC 8236, San Diego, CA 92103, USA; qianjin0027@163.com (J.Q.); d4park@health.ucsd.edu (D.J.P.); sperrott@ucsd.edu (S.P.); PPatel86413@med.lecom.edu (P.P.); 2Department of Plastic Surgery, Shanghai Jiao Tong, University Affiliated Sixth People’s Hospital, Shanghai 200233, China

**Keywords:** extracellular vesicles (EVs), polyvinyl alcohol (PVA) sponge, exosomes, dendritic cells (DCs)

## Abstract

Extracellular vesicles (EVs) have an important role in mediating intercellular signaling in inflammation and affect the kinetics of wound healing, however, an understanding of the mechanisms regulating these responses remains limited. Therefore, we have focused on the use of cutaneous injury models in which to study the biology of EVs on the inflammatory phase of wound healing. For this, the foreign body response using sterile subcutaneous polyvinylalcohol (PVA) sponges is ideally suited for the parallel analysis of immune cells and EVs without the need for tissue dissociation, which would introduce additional variables. We have previously used this model to identify mediators of EV biogenesis, establishing that control of how EVs are made affects their payload and biological activity. These studies in normal mice led us to consider how conditions such as immunodeficiency and obsesity affect the profile of immune cells and EVs in this model using genetically defined mutant mice. Since EVs are intrinsically heterogenous in biological fluids, we have focused our studies on a novel technology, vesicle flow cytometry (vFC) to quantify changes in EVs in mouse models. Here, we show that myeloid-derived immune cells and EVs express proteins relevant in antigen presentation in PVA sponge implants that have distinct profiles in wildtype, immune-deficient (NOD scid) vs. diabetic (Lepr^db^) mice. Together, these results establish a foundation for the parallel analysis of both immune cells and EVs with technologies that begin to address the heterogeneity of intercellular communication in the wound bed.

## 1. Introduction

In the field of cutaneous injury there are a wide range of experimental models where a better understanding of the biological basis of small extracellular vesicles (EVs, 30–120 nm in diameter) in inflammation requires well-defined approaches to characterize the recruitment of immune cells and the release of EVs. Healthy repair of cutaneous wounds is a coordinated response of hemostasis, immune cell recruitment, angiogenesis, and re-epithelialization [1,2]. In contrast, dysregulation of normal cutaneous repair is a significant clinical problem with obesity, diabetes, aging, and infection, placing patients at risk for chronic wounds.

The immune system is a central regulator of inflammation and resolution of tissue repair in skin [3], bone [4], muscle [4], and nerves [5]. The inter-cellular communication that regulates these responses are generally understood to involve direct cell–cell contact-dependent interactions, paracrine factor signaling (i.e., cytokines, chemokines, growth factors), and more recently, EVs [6]. EVs have been proposed to affect immune cells [7,8], angiogenesis [9], fibroblast proliferation [10], and cellular plasticity [11]. In the immune system, the best described function of EVs has been their capacity to trigger inflammatory responses and transport antigen-loaded MHC class I and II complexes [12]. The activity of EVs on antigen-presenting cells includes stimulation of immunogenic responses by EV-mediated presentation of tumor antigens in cancer [13,14], viral antigens in infection [15], and Damage-Associated Molecular Patterns in injury [16]. EVs can also promote tolerogenic and immunosuppressive responses where they modulate cytokine production in monocytes [17], maturation of dendritic cells (DCs) in transplantation, transfer of antigen [18,19,20], delayed onset of inflammation in arthritis [21], and functional transfer of miRNAs between DCs and recipient cells [22]. In various injury models, EVs can activate regulatory T cells, and macrophages, while suppressing pro-inflammatory cytokines and leukocyte adhesion [23,24,25]. The consequence of these EV-mediated changes can lead to increased activity of enzymes involved in tissue remodeling and angiogenesis that enhances cell differentiation and wound repair resolution [26,27,28].

EVs are heterogenous with respect to size and composition, limiting the usefulness of bulk biochemical assays (i.e., Western blot and ELISA), while conventional flow cytometry lacks the specificity and sensitivity to analyze individual EVs. However, single vesicle flow cytometry (vFC) [29,30,31,32,33] provides high resolution EV counting, sizing, and surface protein measurements using membrane-selective dyes, bright fluorescent antibodies, and an appropriately sensitive flow cytometer. Our previous work has demonstrated that by targeting specific biogenesis pathways that regulate how EVs are made and released, it is possible to control the payload and biological activity in wound healing [8]. Here, we sought to unify the most relevant and scientifically rigorous approaches to identify and characterize EVs that are naturally released in genetic models of immunodeficiency and diabetes associated with impaired wound healing. Our workflow incorporates calibration standards and standardized protocols to enable rigorous and reproducible measurements to advance the development of a high-resolution map that addresses the heterogeneity of EVs relevant in tissue repair.

## 2. Results

### 2.1. Immune Cell Recruitment in Immune-Competent Mice

To define cutaneous injury-induced inflammation responses, we focused first on the identification of immune cells and second on the EVs released from cells in the injury site. Subcutaneous implants of PVA sponge were used as a source for both cells and EVs in wound fluid that can be analyzed in parallel without the need for enzymatic dissociation of the tissue. For the study of EV surface markers that may affect the EV tropism and their relevance in immune responses in the wound bed, we quantitate kinetic changes and define the effect of the genetic status of the host on the profile of EVs released (Figure 1A). 

C57 BL/6 mice implanted with PVA sponges were used for the collection and analysis of myeloid immune cells recruited over a 2–14 day time-course, focusing on CD11c^+^ subsets such as macrophages (CD11b^+^MHC^Int^) and antigen-presenting cells (APCs; CD11b^+^MHCII^Hi^) (Figure 1B) [34]. Flow cytometry analysis of cells recruited into the PVA sponge demonstrated an increase in the percentage of total cells identified as CD11b^+^MHC^Int^ macrophages from 10.7% +/− 6.9% at 2 days to 51.1% +/− 7.8% at 14 days (Figure 1C), while the percentage of CD11b^+^MHCII^Hi^ APCs increased from 0.7% +/− 0.7% at 2 days to 12.6% +/− 2.5% at 14 days (Figure 1D). 

Based on the increase in the number of CD11b^+^CD11c^+^ cells identified in the sponge by 14 days, these cells were further evaluated for surface markers of macrophage and/or APC activation (Figure 2). In representative histograms we observed an increase in the number of cells by 14 days (Figure 2A), and in analyses of replicates, levels of CD40, CD86, CCR7, CD135, CD14, and CD117 expressed on the surface of CD11c^+^ immune cells had an increase in the mean fluorescent intensity (MFI) of CD40, CD86, CCR7, CD135, CD14, and CD117 after 14 days (Figure 2B). These findings established the kinetics of immune cell recruitment in the PVA sponge model and identified substantial numbers of CD11b^+^ CD11c^+^ with a DC signature at 14 days and was used as the timepoint used for analyses of other genetic backgrounds relevant in wound healing.

### 2.2. Immune Cell Recruitment Skewed in Mouse Models of Immuno-Deficiency and Diabetes

To determine whether a genetic model of immunodeficiency or diabetic obese mice affected the recruitment of immune cells in a PVA sponge model of cutaneous injury, we first examined non-obese diabetic (NOD) severe combined immune deficient mice that have a spontaneous mutation of the *Prkdc* gene (NOD scid mice) and lack functional T and B cells [35]. Second, we analyzed diabetic obese mice that have a spontaneous mutation for the leptin receptor (Lepr^db^), and become identifiably obese with elevated plasma insulin at 3–4 weeks of age [36]. Using these genetically defined mutant mouse models, which are well known to have impaired wound repair [37,38], we established the profile of immune cell recruitment for subsequent EV analyses, which were compared with the expression of macrophage and APC markers 14 days post-PVA sponge implantation (Figure 3). In representative histograms at 14 days (Figure 3A) and in replicate analyses of CD40, MHCII and CD86 levels (Figure 3B), we compared C57 BL/6 wildtype (B6) to mutant mice and observed decreases in the levels of CD40 and CD86 on CD11c^+^ cells in both NOD scid and Lepr^db^ (DB) mice. However, MHCII levels were decreased significantly only in NOD scid, while changes in MHCII in Lepr^db^ mice were not significant, likely due to high variability (Figure 3B, Middle). Analysis of other markers on CD11c+ cells (Figure 3C) showed increased levels of the hyaluronic acid receptor CD44 and macrophage marker F4/80 in NOD scid mice (*p* < 0.03). Interestingly, the levels of granulocyte marker Gr1 was increased in both NOD scid and Lepr^db^ mice, while no changes were observed in Ly6G levels. These results demonstrated that the genetic status of the host affected the profile of immune cells recruited in a wound bed and provided the context for the analysis of EVs in this model. 

### 2.3. Analysis of EVs in a Defined Injury Model of Immune Cell Recruitment 

Based on the workflow above outlining the collection of EVs in parallel with cells (Figure 1A), we enriched EVs by serial centrifugation as shown (Figure 4). EVs were subjected to labeling for subsequent vFC (Figure 5A), nanoparticle tracking analysis (NTA) established a mean diameter of EVs as 117 +/− 1.9 nm, consistent with their identification as EVs (Figure 5B). As detailed in the Materials and Methods and Figure 4, enriched EVs were isolated by ultracentrifugation, the amount normalized by protein concentration, and subjected to labeling with the lipophilic membrane dye Di8-ANEPPS that undergoes an increase in fluorescence upon binding in the lipid bilayer [30]. To demonstrate the specificity of Di-8 ANEPPS labeling of EVs, a serial dilution of the dye was performed (Figure 5C, Top). The fraction of detergent soluble EVs was determined by treatment with Triton X-100 (Figure 5C, Bottom), thus identifying detergent-sensitive EVs in a flow cytometry dot plot of Di8-ANEPPS labeled samples (Figure 5C, Top). The protein concentration increased from 2–14 days (Figure 5D), establishing a standard for normalizing the input for subsequent vFC assays of EVs at 10 µg per assay. Immunoblotting of 14 day EVs confirmed the expression of EV marker proteins ALIX, CD9, CD63 and CD81 (Figure 5E) [39]. Focusing on the expression of genes known to have the potential to regulate EV biogenesis (i.e., *Rab5A, Rab35, TSG101,* and *VPS4*), we isolated total cells from the PVA sponge at 2 and 14 days and observed a significant increase in *Rab5A* (Figure 5F). These studies establish the methods for the characterization of EVs and the expression of a subset of genes that regulate EV biogenesis in the PVA model.

### 2.4. Kinetics of EV Protein Changes in PVA Sponges

To quantitate kinetic changes in EV protein levels, vFC was performed with antibodies directed to tetraspanins, integrins, and markers of immune cell activation (Table 1). EVs labeled with Di8-ANEPPS (Figure 6A, Left) were then labeled with fluorescent antibodies (Figure 6A, Middle), and compared with no antibody as a sham control (Figure 6A, Right). Using this approach to quantify EV proteins, we analyzed PVA sponge EVs isolated 2–14 days post-implantation. The number of EVs expressing CD9 and CD18 were significantly decreased (*p* < 0.05) by 14 days, while CD103, CD29 and MHCII were increased by day 14 (*p* < 0.05) (Figure 6B). There were no significant changes between 2 and 14 days in the levels of CD63, CD117, CD11c, CD64 and IRF4. These studies quantify the temporal regulation of EV protein expression in the wound bed.

### 2.5. Changes in Wound EV Proteins in the Wound Bed Immunodeficient and Diabetic Mice

To determine whether genetic models of immunodeficiency or diabetes affect the profile of EVs released in a wound bed, we used the PVA model to analyze changes in EV protein levels in wildtype C57 BL/6, NOD scid, and Lepr^db^ mice. EVs from NOD scid mice expressed reduced levels of tetraspanins CD9 and CD63 (*p* < 0.05) and DC/macrophage related markers MHCII and CD64 (*p* < 0.05) compared to immune-competent C57 Bl/6 mice. However, IRF4 was increased in NOD scid mice, with no significant changes detected for integrins CD117, CD103, CD29, CD18 and CD11c (Figure 7A–J). In Lepr^db^ mice, higher levels of CD9, CD63, CD117, CD103, CD29, CD18, CD11c, CD64, and IRF4 were expressed compared to wildtype mice, but no significant change in MHCII (Figure 7A–J). The combination of quantitative approaches for the measurement of cells (Figure 1, Figure 2 and Figure 3) and EV proteins (Figure 4, Figure 5, Figure 6 and Figure 7) established a foundation for the analysis of relevant immune mediators in the wound bed of genetically defined animal models.

## 3. Discussion

Clinical observations and experimental studies of tissue repair hold that the resolution of injury comprises four main phases: hemostasis, inflammation, proliferation, and remodeling1, with EVs being potent mediators of the inflammation phase [40,41,42,43]. We and others have shown that EVs are carriers of protein and nucleic acids, with their profile and payload affecting the biology of tissue repair [44,45,46]. Our over-arching hypothesis is that the classical phases of tissue repair have distinct subsets of EVs associated with them that can be regulated in terms of profile and payload to modify the inflammation phase, leukocyte recruitment and resolution of tissue injury. Along with full thickness cutaneous excisional wounds, implants of polyvinyl alcohol (PVA) sponge as a foreign body response that promotes the recruitment of immune cells. PVA implant models in particular enable a side-by-side immunophenotyping of immune cells and EVs, since both cells and EVs are easily recovered for analysis without tissue digestion. The PVA sponge model is also an ideal donor site since, as a 3-dimensional scaffold, it supports: (a) a higher density of leukocyte-derived EVs than is possible in cell culture, while also avoiding serum and tissue culture plastic (b) all cells are primary and can be mapped to equivalent cell types in full thickness wounds in mouse and human models, (c) mimics many of the technical advantages of a ‘bioreactor approach’ [47] in which EVs are collected from higher density cells in PVA sponges, and are easier to recover since there is minimal handling/dissociation, and (d) ease with which PVA can be modified by coating with various ECM proteins to select for EVs with a particular tropism. We have recently published on the characterization of PVA sponge studies to show that adoptive transfer of EVs collected from PVA sponges in donor mice can be adoptively transferred to naïve recipient mice where they mediate the recruitment of immune cells and wound closure [8]. Here, parallel analyses of immunocompetent, immunodeficient, and immunodeficient mice engrafted with donor cells and/or EVs will help define inflammation responses relevant to tissue repair [48]. In the PVA sponge model, cells and EVs can be recovered in tandem from the wound bed without the need for tissue dissociation and can be applied to the study of genetically defined mouse models of diabetes and immunodeficiency. With the PVA sponge model being devoid of resident cells, the immune cells recruited into the sponge are primarily from the circulation, along with stromal cells that infiltrate the implant. Here, we demonstrate, using flow cytometry, that both macrophage and DCs are recruited into the sponge implant. The profile and kinetics of these myeloid cell populations are distinct in wildtype, diabetic (i.e., Lepr^db^) and immune-deficient mice (NOD scid). Similarly, the profile of EVs in the wound fluid of the PVA sponge implant was defined in the terms of the kinetics and genetic status of the host. Changes in the EV profile are quantified by vFC, providing one of the most comprehensive in vivo EV analyses in a wound model to date.

Analyses of infiltrating immune cells demonstrated an increase in markers of antigen presentation such as MHCII, CD40, CD86, CCR7 consistent with the recruitment and/or differentiation of myeloid immune cells capable of adaptive immune responses. Interestingly markers of hematopoietic progenitor cells CD117 and CD135 were also increased, indicating that the PVA sponge may support a niche of stem cells, or marker of immature monocytes/macrophages in the wound bed that increased from relatively low levels at 2 days to high levels by 14 days. These kinetic studies established the 14 day timepoint as a focus for comparisons with genetically defined models. Here, changes in antigen presentation markers were consistently decreased in NOD scid mice, which we attribute to lack of T and B cells and the absence of feedback adaptive responses such as DC activation are low when T cells are absent. Interestingly, in Lepr^db^ mice, MHCII levels were unchanged compared to wildtype C57 BL/6 mice, which might be related to the absence of additional stressors that would contribute to a chronic phenotype. Levels of other markers of macrophages (i.e., F4/80), and granulocytes (i.e., Gr1 and Ly6G) were relatively consistent in NOD scid and Lepr^db^ mice. In the design of the model, the parallel analysis of cells and EVs in a complex subcutaneous microenvironment was critical to identify correlations between cells and EVs released that are most relevant in a wound bed. Furthermore, vFC assays that address the complexity of EVs when used in parallel with standard flow cytometry to identify immune cells. Therefore, we first established EV size, detergent solubility, concentration, and expression of tetraspanins, along with a small survey of some of the most widely cited EV biogenesis genes. Second, we measured the levels of proteins, selected based on their relevance in immune responses, on individual EVs using vFC. We identified both kinetic changes in these EV proteins as well as differences between wildtype C57 BL/6, NOD scid and Lepr^db^ mice. Interestingly for some proteins central to antigen presentation like MHCII, changes in cells vs. EVs were similar, with decreases in NOD scid but not Lepr^db^ mice. For other proteins that were detectable on cells such as CD40 and CD86, there was no significant detectable levels on EVs. However, we did identify EV proteins that had selective increases in Lepr^db^ mice compared to C57 BL/6 such as CD9, CD63, CD117, CD103, CD29, CD11c, MHCII, CD64, and IRF4. Interestingly, levels of CD18 (integrin β2) were highly increased in Lepr^db^ mice. Overall, while the differences in protein levels on EVs between wildtype mice and NOD scid were relatively small, most were statistically significant in small sample sizes. We anticipate that these findings can be used as a foundation to better understand how changes in markers of APC activation on cells contrast with the expression of these markers on EVs drive local vs. distal immune cell signaling (i.e., wound bed vs. skin draining lymph nodes). Another level of EV-mediated signaling may occur in normal immune-competent animals vs. immune deficient or diabetic animals. Differential capacity for EVs in these models to support APC activation may provide insights into the mechanisms driving the resolution of inflammation in chronic wounds.

## 4. Materials and Methods

### 4.1. Mice and PVA Sponge Implant and Tissue Harvest

For the preparation and analysis of EVs in this study, we utilized standards published by The International Society for Extracellular Vesicles for EV enrichment (MISEV guidelines) [39] and vFC (MIFlowCyt-EV guidelines) [49], respectively. All mouse procedures were approved by the University of California in San Diego Institutional Animal Care and Use Committee (Protocol #S07339, Approved 5 February 2021). All studies were performed with a minimum of 5 mice, each with 3 sponge implants to establish a sample size with sufficient statistical power to the following cellular and biochemical analyses. C57 BL/6 (#000664), NOD scid (#001303) and Lepr^db^ mice (#000697) (male, 8–10 weeks of age) were kept at room temperature with a 12-h light/dark cycle, and subjected to implantation of sterile polyvinyl alcohol (PVA) sponges (PVA Unlimited, Warsaw, IN, USA) into the dorsum, and the wound site closed with three sutures. Animals recovered in the presence of sufficient food and water supply. At indicated post-operative days 2, 7 or 14, mice were euthanized, and the sponges removed, placed in a Petri dish with 1 mL of PBS. Cells were separated from EV-containing supernatant by centrifugation at 535× *g* for 5 min. The supernatants were subjected to further enrichment of EVs as described below while cell pellets were resuspended in PBS and subjected to RNA isolation and flow cytometry analysis as described below.

### 4.2. Enrichment of Extracellular Vesicles from PVA Wound Fluid

To enrich EVs from the supernatant of PVA sponge wound fluid, centrifugation was performed at 10,000× *g* for 30 min at 4 °C, and the supernatant collected and subjected to ultracentrifugation at 100,000× *g* at 4 °C for 70 min in a Beckman Optima Max-XP Ultracentrifuge with a TLA120.2 rotor (clearance factor = 39) (Beckman Coulter, Brea, CA, USA). Additional details are provided in the Appendix A. Supernatants were then removed and discarded while the pellets containing EVs were resuspended in 100 µL PBS for each sample and used for subsequent protein concentration, NTA, immunoblot and vFC analyses. To prepare protein lysates of EVs for determination of protein concentration and immunoblotting, EVs were solubilized in detergent containing RIPA lysis buffer (#89901, Thermo Fisher Scientific, Waltham, MA, USA), and the protein concentration measured with a BCA assay kit (#23227, Thermo Fisher Scientific) as indicated by the manufacturer.

### 4.3. Flow Cytometry Analysis of Cells

Cells harvested from PVA sponges were subjected to flow cytometry to detect infiltrating immune cells by incubation with Fc block (Cat#130-092-575, Miltenyi Biotec, San Diego, CA, USA), followed by staining with antibodies specific for the following immune cell markers from Biolegend (MHCII #107651; CD11c #117363; CD11b #101223; CD40 #124609; CD86 #105011; CD135 #135305; CD14 #12331; CCR7 #150627; and CD117 #105807), and Miltenyi Biotec (CD11b #130-109-288; CD45 #130-110-803; CD44 #130-110-085; F4/80 #130-102-422; Gr1 #130-102-233; and Ly6G #130-107-912, propidium iodide (130-093-233, Miltenyi Biotec) to exclude dead cells, and fluorescence minus one (FMO) used a controls for antibody specificity. All flow cytometry was performed on a MACSQuant 10 instrument (Miltenyi Biotec) and analyzed using FlowJo software (Version 10.7.1, Becton Dickinson, Franklin Lakes, NJ, USA).

### 4.4. Analysis of EVs by Immunoblotting 

EVs enriched from PVA sponges were suspended in PBS, solubilized in RIPA, heated to 95 °C in NuPAGE^®^ LDS Sample Buffer (NP0008, Invitrogen, Carlsbad, CA, USA) for 5 min, fractionated in precast 12% acrylamide gels, and transferred to PVDF membranes. Following blocking in 3% nonfat dry milk (NFDM), the following primary antibodies were added in NFDM at 4 °C for overnight incubation: CD9 (MA5-31980, 1:1000, Invitrogen, Carlsbad, CA, USA), CD63 (MA5-92370, 1:1000, Invitrogen, Carlsbad, CA, USA), CD81 (#10037, Cell Signaling Technologies, Boston, MA, USA), or Alix (#92880, 1:000, Cell Signaling Technologies). Anti-rabbit IgG HRP-linked antibody (#7074, 1:1000, Cell Signaling Technologies) was used to bind primary antibodies, and detected with enhanced chemiluminescent substrate and imaged with Lumina cooled CCD camera system (Perkin Elmer, Concord, MA, USA).

### 4.5. Nanoparticle Tracking Analysis (NTA)

The particle concentration, size, distribution of the isolated PVA-derived EVs was analyzed by a NanoSight^®^ NS300 (Spectris, Egham, UK). For particle visualization and recording of light scattering, 1 mL of a 1:100 dilution of EVs were analyzed in three 60 s video recordings with a detection threshold of 5, blur size control set to automatic, and analyzed for EV concentration and size distribution (Acquisitions software, NTA 3.1, MalvernPananalytical Ltd., Malvern, UK).

### 4.6. Quantitative Real-Time PCR

Real-time PCR (qRT-PCR) was performed on cDNA prepared from 2 μg of total RNA using (Thermo Fisher Scientific, San Diego, CA, USA) reverse transcriptase according to manufacturer’s directions (iScript, #1708891, Biorad, Hercules, CA). Gene expression was measured using SYBR^®^ Green (# 1725271, Biorad) on a CFX96 Touch™ Real-Time PCR Instrument with Quantitect^®^ primers from Qiagen (Venlo, The Netherlands) specific for the following genes: *Rab5a* (QT00171969), *Rab35* (QT00107268), *Vps4a* (QT00145089), and *Tsg101* (QT00143192). qRT-PCR was performed with the following parameters: a preamplification at 95 °C for three min, 40 cycles of 95 °C for 10 s, 55 °C for 10 s, 72 °C for 30 s for a melt curve analysis. All processes were performed in duplicate, and analyzed by CFX manager software (Version 3.1, Biorad, Hercules, CA, USA).

### 4.7. Vesicle Flow Cytometry

Isolated EVs were subjected to vesicle flow cytometry (vFC) using a specific fluorescent membrane dye Di8-ANEPPS (D3167, Thermo-Fisher) as previously described30. Briefly, EV samples were labeled with Di8-ANEPPS staining buffer (12 μM) supplemented with Pluronic F-127(#P6866, Thermo-Fisher) at room temperature for 30 min and then stained with following APC-conjugated antibodies listed in Table 1. Configuration of the MACSQuant10 (MQ10) was set to trigger on events over 1.1 V on the basis of Di8-ANEPPS (B3 channel on the MQ10 to focus analysis on dye-labeled events. Additional details are provided in the Appendix A. To identify detergent soluble EVs and to avoid ‘swarm’ and other background artifacts, samples were analyzed twice, first without detergent and then second after the additive of the detergent Triton X-100 (T8532, Sigma Aldrich, St. Louis, MO, USA). This sequential strategy enables the selection of a gate that identifies detergent soluble Di8-ANEPPS positive EVs from which additional staining with high index fluorophore such as APC can be used for the detection of antibodies bound to EVs.

### 4.8. Statistical Analysis 

All statistical analyses were performed with Prism 6.0 (Graph pad Software, La Jolla, CA, USA). Descriptive results of continuous variables were expressed as the mean ± standard deviation for normally distributed variables. Differences between different groups were compared by Student’s *t* test *p*-values less than 0.05 were considered to be statistically significant.

## 5. Conclusions

Overall, these studies provide the basis to compare immune cell recruitment with EV release in the same animals with minimal handling. We anticipate that these studies can provide the foundation for functional uptake studies of EV payloads in animal models and patient samples that may be useful for diagnostic and therapeutic applications.

## Figures and Tables

**Figure 1 ijms-22-03551-f001:**
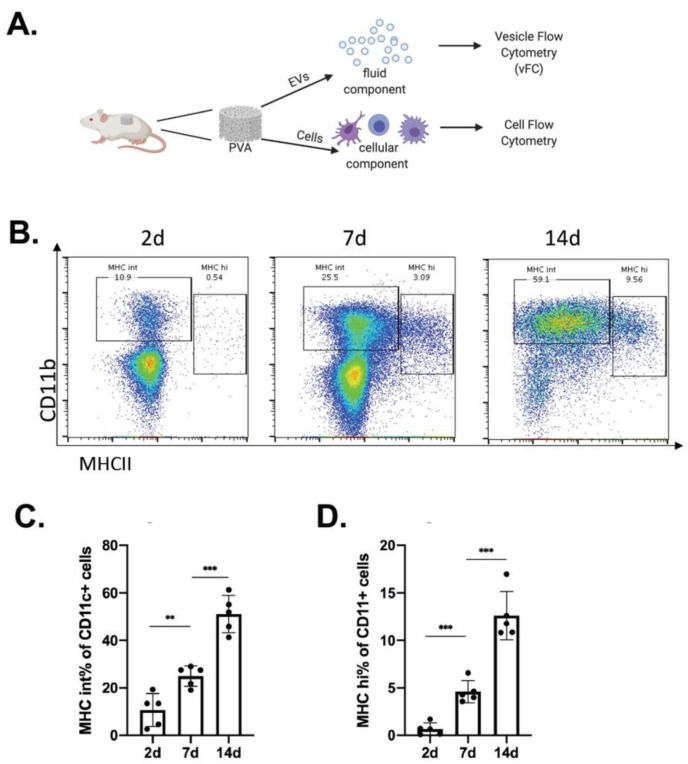
Kinetics of immune cell recruitment in the in vivo foreign body response using immune competent mice. (**A**) Subcutaneous implants of PVA sponge were used as a model of sterile inflammation were immune cells were isolated and analyzed by flow cytometry to define cell types, and the wound fluid used for subsequent EV analysis. (**B**) To focus on myeloid-derived immune cells, representative dot plots of events that were first gated on CD11c^+^ cells to identify the relevant subset of cells to analyze, and the kinetics changes in surface levels of CD11b and MHCII determined at 2, 7, and 14 days post-implantation. (**C**) Quantification of changes in the numbers of CD11c^+^ cells expressing intermediate levels of MHCII (MHCIIint), (**D**) changes in cells expressing high levels of MHCII (MHCII^hi^). Data shown are mean ± SD. (*n* = 5; ** *p* < 0.01 and *** *p* < 0.001).

**Figure 2 ijms-22-03551-f002:**
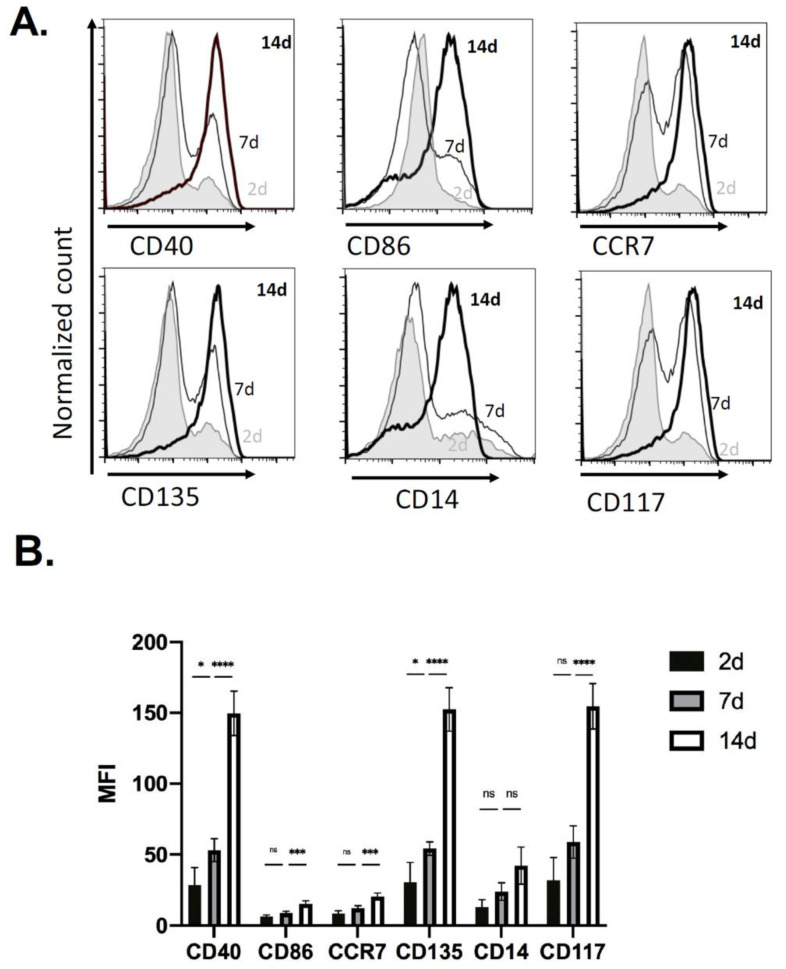
Expression of dendritic cell and hematopoietic stem cell markers in CD11c+ cells in PVA sponge implants. (**A**) Expression of CD40, CD86, CCR7, CD135, CD14, and CD117 on cells recruited to PVA sponges analyzed 2, 7 and 14 days post-implantation. (2 day, shaded; 7 day, thin line; 14 day, bold line). (**B**) Quantification of protein expression on recruited cells (2 day, black shading; 7 day, gray shading; 14 day, open). Data shown are mean ± SD (*n* = 5; * *p* < 0.05, *** *p* < 0.001, **** *p* < 0.0001, ns = not significant).

**Figure 3 ijms-22-03551-f003:**
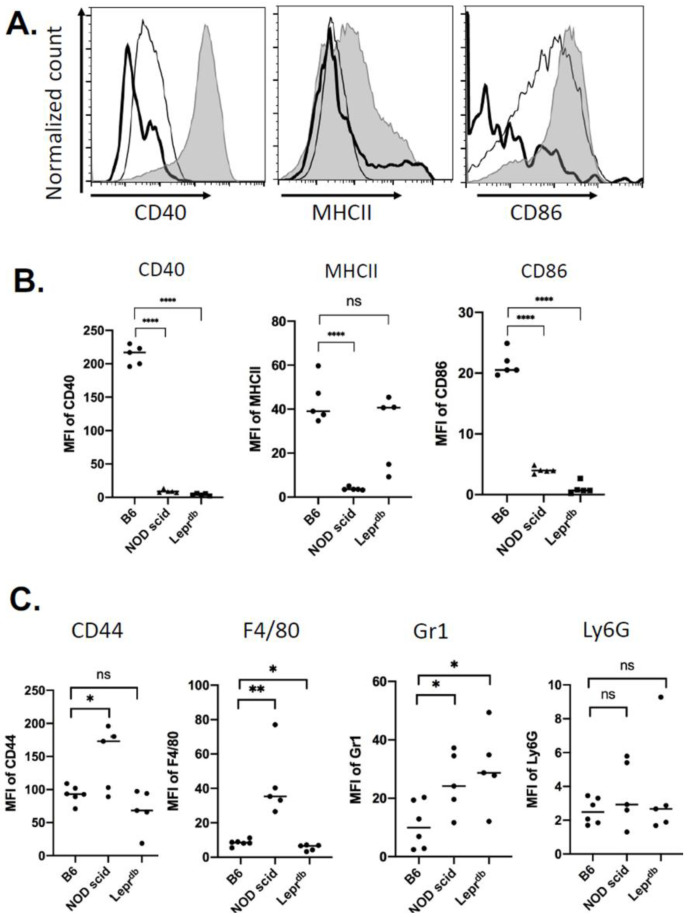
Immunophenotype of CD45^+^CD11c^+^ immune cells recruited in wild type, immunodeficient, and diabetic-obese mice implanted with PVA sponges. (**A**) Expression of CD40, MHCII, and CD86 in wildtype C57 BL/6 mice (shaded), immunodeficient NOD-scid mice (thin line), and diabetic-obese Lepr^db^ mice (bold line). (**B**) Quantification in replicate mice (*n* = 3), and (**C**) analysis of macrophage and granulocyte markers (*n* = 5; * *p* < 0.05, ** *p* < 0.01, and **** *p* < 0.0001, ns = not significant).

**Figure 4 ijms-22-03551-f004:**
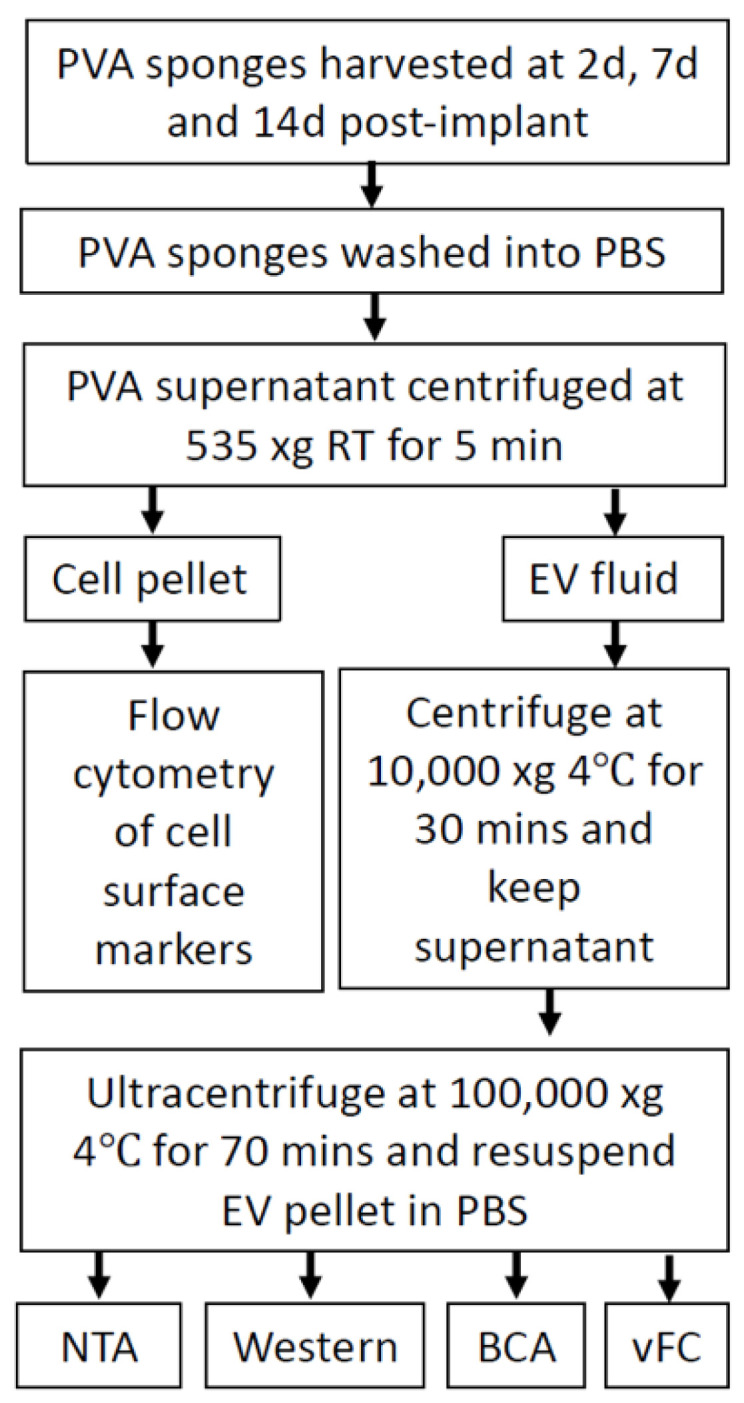
Overview of enrichment of extracellular vesicles released in PVA sponge implants. Wound fluid harvested from PVA sponges was used for analysis of cells and EVs by serial centrifugation.

**Figure 5 ijms-22-03551-f005:**
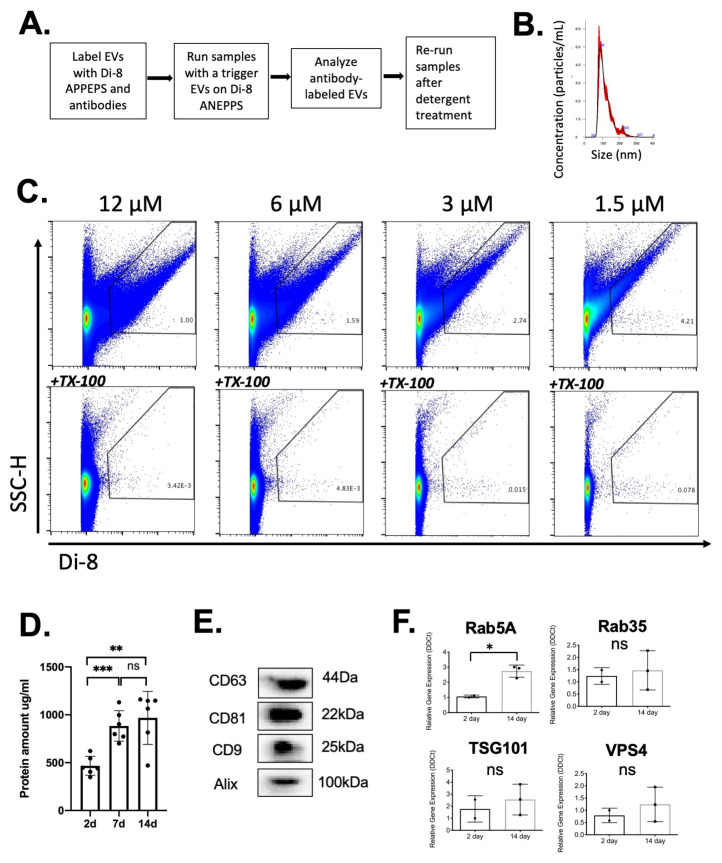
Characterization of EVs released in the PVA sponge model in C57 BL/6 mice. (**A**) Summary of vFC analysis. (**B**) NTA analysis of size distribution of EVs following ultra-centrifugation. (**C**) Analysis of EVs labeled with di-8 ANEPPS (top row), and gating on Triton-X100 detergent sensitive EVs (bottom row). (**D**) Quantification of protein concentration at 2, 7 and 14 days post-implantation used to normalize vFC analysis of EVs. Data shown are mean ± SD (*n* = 5; * *p* < 0.05, ** *p* < 0.01, and *** *p* < 0.001). (**E**) Immunoblotting of EVs to detect Alix, CD9, CD63, and CD81 at 14 days post-implantation of PVA sponges. (**F**) The gene expression analysis of genes associated with EV biogenesis in PVA cells recruited to PVA sponge at 2 and 14 days (* *p* < 0.05, ns = not significant).

**Figure 6 ijms-22-03551-f006:**
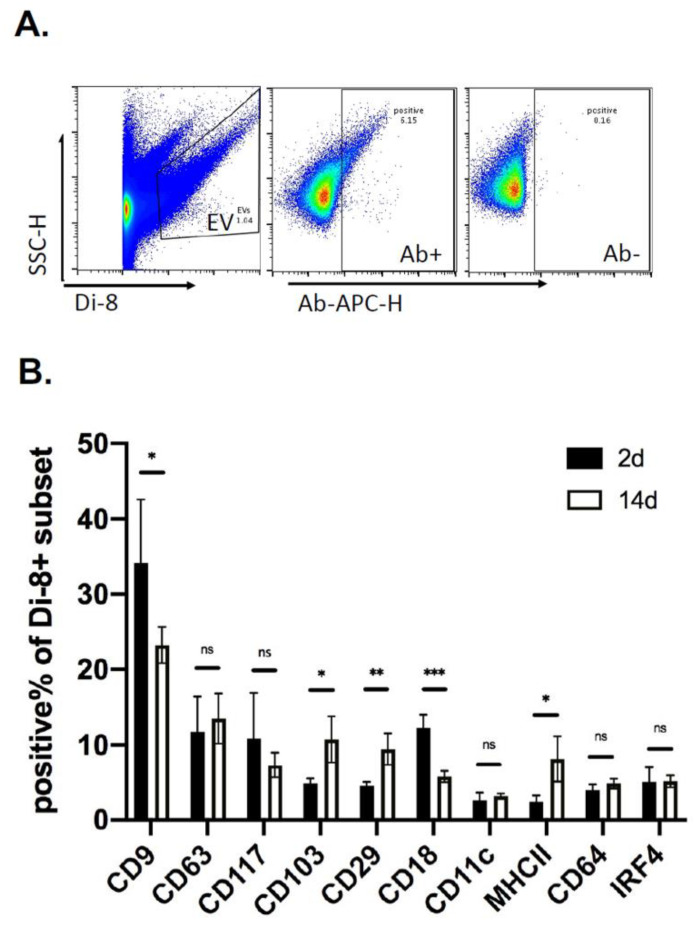
Kinetics of EV proteins in PVA sponges measured using vFC. (**A**) EVs isolated from C57 BL/6 mice were normalized by protein concentration, stained with di-8 fluorescent dye, labeled with allophycocyanin (APC)-labeled antibodies, and analyzed by modified flow cytometry as described in the Materials and Methods. (**B**) Expression of immune cell-related proteins on the surface EVs analyzed 2 and 14 days post-implantation. Data shown are mean ± SD. (*n* = 5; * *p* < 0.05, ** *p* < 0.01, and *** *p* < 0.001).

**Figure 7 ijms-22-03551-f007:**
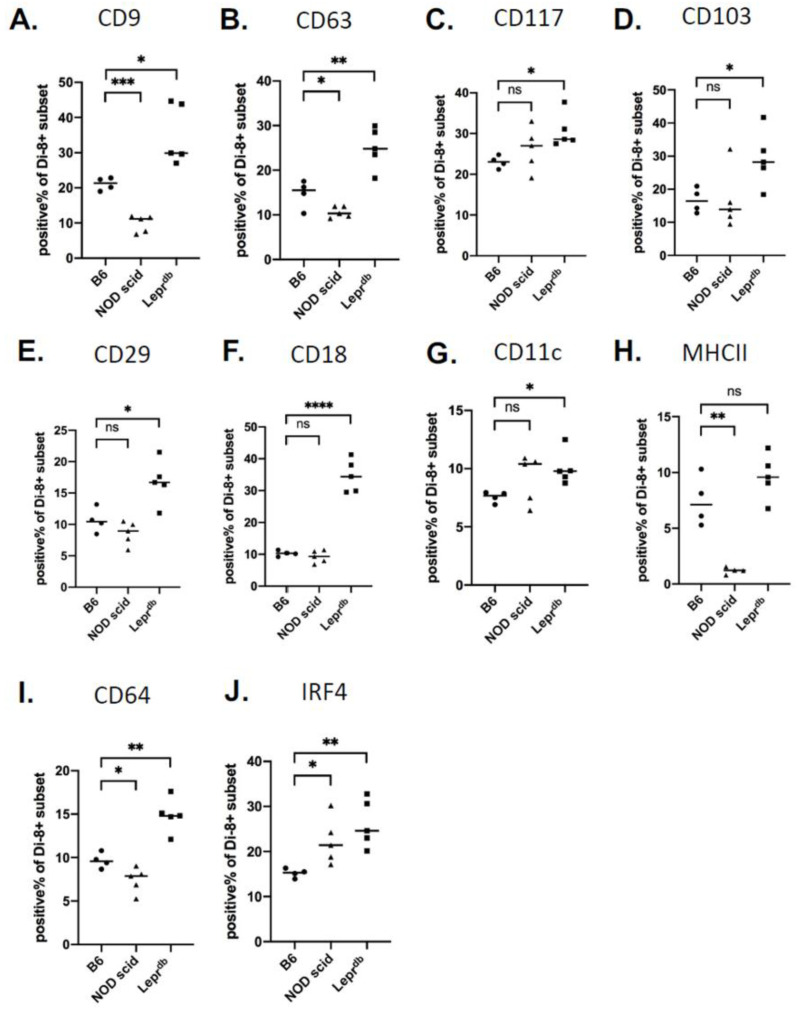
Analysis of EV proteins in wildtype, immunodeficient, and diabetic-obese mice. (**A**–**J**) Protein expression on EVs isolated and analyzed by vFC on C57 BL/6, NOD scid, and Lepr^db^ (DB) mice. Data shown are mean ± SD. (*n* = 4 for C57 BL/6 and *n* = 5 for NOD scid and DB mice group; * *p* < 0.05, ** *p* < 0.01, *** *p* < 0.001, and **** *p* < 0.0001, ns = not significant.

**Table 1 ijms-22-03551-t001:** Antibodies used for analysis of EV surface markers using vesicle flow cytometry.

Antibody	Clone	Flour	Cat#	Vendor
CD9	MZ3	APC	130-102-612	Miltenyi Biotec
CD63	REA563	APC	130-108-924	Miltenyi Biotec
CD103	2E7	APC	130-102-516	Miltenyi Biotec
CD117	3C11	APC	130-122-948	Miltenyi Biotec
CD29	HMβ1-1	APC	130-102-557	Miltenyi Biotec
CD18	M18/2	APC	130-104-019	Miltenyi Biotec
CD11c	N418	APC	130-119-802	Miltenyi Biotec
MHCII	M5/114.15.2	APC	130-123-785	Miltenyi Biotec
CD64	X54-5/7.1	APC	139305	Biolegend
IRF4	IRF4.3E4	Alexa Flour 647	646407	Biolegend

## Data Availability

Available by contacting the corresponding author.

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
