# Peer review of "Genetic Background and Kinetics Define Wound Bed Extracellular Vesicles in a Mouse Model of Cutaneous Injury"

_ijms, 2021, doi:10.3390/ijms22073551_

Round 1
Reviewer 1 Report
In their work Qian et al describe the tropism and immunomodulatory potential of extracellular vesicles in the wound bed. To study the different immune infiltrating cells and their respective EVs they compare 3 mouse models with different genetic background: C57BL/6 wildtype, NOD scid and Leprdb. According to the authors this allows investigating cell recruitment in wound bed in immunodeficiency and diabetes settings.
The work is interesting and the evaluation of the EVs in wound bed using the different models may have important implications also for other settings like cancer. The employed methods are accurate and the results are interesting. The figures in the article are difficult to read, although the .tif versions are ok. There are some issues that should be addressed.
The abstract should summarize the findings and their meaning, not only what has been done. It is not clear where the new work starts with respect to the already published work. Please change the abstract accordingly.
In the introduction the previous findings (reference 8) from the same authors and representing the basis of this new work should be described. In its current form the reader cannot understand what the new work adds to the already published one without reading it, however a brief summary of the previous findings would enhance the relevance of the results presented here.
The rational for this specific research should clearly emerge at the beginning and also in the abstract, i.e. why was this work done? Why were these models chosen and what do the results tell us, what implications do they have for immunodeficient and diabetic patients?
In the immune competent model the authors should give information about all immune cells/EVs from these immune cells present in the wound bed, also the knowledge about their absence is important to get a complete overview to then compare with the other models.
Did the authors check for differences in NTA between the 3 models? Was there a correlation of CD9 and EV measurements by NTA?
From which model do EVs derive shown in Figure 5? The characterization is shown only for one model, were EVs from the other 2 models different? Did the authors ever perform electronmicroscopy to visualize their EVs? The EV characterization should be implemented with the data from the other 2 models.
Figure 5 F: was the expression of genes associated with EV biogenesis here shown in the PVA recruited cells different among the models, as shown for the cells in figure 3A?
At the end of the discussion please add some interpretation of the observed and described phenomena: what do according to authors the changes going together in cells and EVS (MHCII) mean, while other proteins change on the cells but not on the EVs.
Minor points:
The centrifugation is 461 g on page 2 but in Figure 4 it is 535 g.
The list of used Antibodies can be added to the supplementary information.
Please add IgG controls for cell and EV flow cytometry. In the cell flow cytometry: did the cells already express all markers? Were there differences between the models
Author Response
We thank the reviewers for their thoughtful and helpful questions that have improved the manuscript significantly. We decided that based on these constructive questions, a re-write of the abstract was appropriate to address some of the over-arching questions.
Q1. The abstract should summarize the findings and their meaning, not only what has been done. It is not clear where the new work starts with respect to the already published work. Please change the abstract accordingly.
Response: Thank you for your comment. We have re-written the abstract.
Q2. In the introduction the previous findings (reference 8) from the same authors and representing the basis of this new work should be described. In its current form the reader cannot understand what the new work adds to the already published one without reading it, however a brief summary of the previous findings would enhance the relevance of the results presented here.
Response: We have clarified the novelty of this study, and that our previous work in FASEB Journal focused on EV biogenesis mechanisms whereas this paper addresses the effect of the host strain on EVs.
Q3. The rational for this specific research should clearly emerge at the beginning and also in the abstract, i.e. why was this work done? Why were these models chosen and what do the results tell us, what implications do they have for immunodeficient and diabetic patients?
Response: We have edited the Abstract and Introduction to address these comments. Briefly, this work is essential because the PVA sponge model is the only one to date that allows for the collection and analysis of both immune cells and EVs from a wound bed over an extended time course (i.e 14 days), without the need for tissue dissociation. Edits to the Discussion address relevance to patients.
Q4. In the immune competent model the authors should give information about all immune cells/EVs from these immune cells present in the wound bed, also the knowledge about their absence is important to get a complete overview to then compare with the other models.
Response: We are interested in identifying EVs released by infiltrating myeloid cell types (i.e. monocytes and macrophages) that may act upon antigen presenting cells (APCs), and possibly function in distal sites like skin draining lymph nodes. Therefore, we have focused our analyses on myeloid cell types and APC activation markers on immune cell in Figures 1-3. We would agree that the study of additional immune cell types (i.e. lymphocytes, NK cells, etc) would be interesting, but these cell types are relatively rare in a sterile model. Therefore, we are currently exploring modifications of the model (i.e. sterile vs. infection) on the recruitment and local activation of lymphocytes in future studies.
Q5. Did the authors check for differences in NTA between the 3 models? Was there a correlation of CD9 and EV measurements by NTA? From which model do EVs derive shown in Figure 5? The characterization is shown only for one model, were EVs from the other 2 models different? Did the authors ever perform electron microscopy to visualize their EVs? The EV characterization should be implemented with the data from the other 2 models.
Response: Figure 7 compares EVs from all 3 genetic models using a vFC as the quantitative technique since it is the only technology to date that addresses the heterogeneity of EVs. Since the wound bed comprises multiple EV subsets, in our opinion vFC is best suited to measure EV heterogeneity since it uses a flow cytometer to examine individual particles. In contrast batch techniques such as NTA and Western Blotting have limited utility for our interests, and thus are used only as baseline techniques to introduce our technology using C57Bl/6 mice (i.e. Figure 5). CD9 is indeed an interesting tetraspanin which is easily detected by immunoblot (Figure 5E) and vFC (Figure 6B), but we did not attempt to correlate CD9 with NTA, since Western blots are not generally quantitative and also limited as a batch technique like NTA as discussed above. We do not perform EM since it is not quantitative. We argue that the comparison of EVs in different models is best accomplished by vFC since it is both quantitative and capable of analysing EVs on an individual basis (and therefore most relevant in a complex biological fluid).
Q6. Figure 5 F: was the expression of genes associated with EV biogenesis here shown in the PVA recruited cells different among the models, as shown for the cells in figure 3A?
Response: This was only tested in C57Bl/6 mice. We agree it would be interesting to perform a more comprehensive gene analysis in various mouse models, but to be informative at a cellular level this would need to be cell type specific study utilizing a technique such as scRNAseq, which is beyond the scope of this study.
Q7. At the end of the discussion please add some interpretation of the observed and described phenomena: what do according to authors the changes going together in cells and EVS (MHCII) mean, while other proteins change on the cells but not on the EVs.
Response: We have edited the discussion to address these excellent points.
Minor points:
Q8. The centrifugation is 461 g on page 2 but in Figure 4 it is 535 g.
Response: This has been corrected.
Q9. The list of used Antibodies can be added to the supplementary information.
Response: Thank you for your comment. We have opted to feature this information prominently since vFC is dependent on antibody specificity and the use of high index fluorophores.
Q10. Please add IgG controls for cell and EV flow cytometry. In the cell flow cytometry: did the cells already express all markers? Were there differences between the models
Response: Our cell and EV flow cytometry uses fluorescence minus one (FMO) controls, a standard approach in flow cytometry (Curr Protocols Flow Cytometry, 2002, PMID 18770762). Yes, the EV markers used were selected on the basis of their detection on PVA sponge cells, however, for EV studies we focused on markers that changed on cells over 2-14 days or on EVs between the three strains, in the interest of space and our focus on tetraspanins, integrins, and APC activation markers.
Reviewer 2 Report
The manuscript by Qian et al describes the most relevant and scientifically rigorous approaches to identify and characterize EVs released in genetic models of immunodeficiency and diabetes. The authors have used vesicle flow cytometry to characterize the EVs. This can overcome the limitations of conventional flow cytometry.
1.The authors suggested that inflammation is one of the phases of tissue repair. Cytokines play a major role in the inflammation pathway and EVs are known to package cytokines and transport them to other tissues leading to various biological effects. The study findings can be further strengthened by measuring cytokines in the EVs and compare between different study groups.
2. Few reference numbers have not been mentioned in (). please check.
Author Response
We thank the reviewer for their comments and have attached responses below.
Q1. The authors suggested that inflammation is one of the phases of tissue repair. Cytokines play a major role in the inflammation pathway and EVs are known to package cytokines and transport them to other tissues leading to various biological effects. The study findings can be further strengthened by measuring cytokines in the EVs and compare between different study groups.
Response: Thank you for your comment and suggestion.
Q2. Few reference numbers have not been mentioned in (). please check.
Response: Thank you for your comment. We proof-read and corrected the references cited.
Round 2
Reviewer 1 Report
Even if I do not fully agree with the views of the authors regarding the issues I raised, the answers they provided are reasonable and the manuscript has now improved. I believe that the results presented in this work are of substantial interest to the audience especially to the EV field scientists.
Reviewer 2 Report
The authors have adequately answered the comments.